# Performance evaluation of Biofire Film Array Respiratory Panel 2.1 for SARS-CoV-2 detection in a pediatric hospital setting

**Mirta Mesquita Ramirez**[1], **Miria Noemi Zarate**[1], **Leonidas Adelaida Rodriguez**[1], **Victor Hugo Aquino**[2]*

1 General Pediatric Hospital "Children of Acosta Ñu," San Lorenzo, Central, Paraguay, 2 Immunology Department, Research Institute for Health Sciences, National University of Asuncion, San Lorenzo, Central, Paraguay

* vhaquino@iics.una.py

## Abstract

The highly contagious nature of the severe acute respiratory syndrome coronavirus 2 (SARS-CoV-2), the causative agent of Coronavirus Disease 2019 (COVID-19), requires rapid diagnostic tests to prevent the virus from spreading within hospitals and communities. Reverse transcription followed by the polymerase chain reaction (RT-PCR) test is the gold standard for detecting SARS-CoV-2 infections but is time-consuming, labor-intensive, and restricted to centralized laboratories. There is a growing need to develop and implement point-of-care and rapid tests for SARS-CoV-2 detection to address these limitations. We aimed to evaluate the performance of BioFire Film Array Respiratory Panel 2.1 (BioFire FA-RP2.1) for SARS-CoV-2 detection in a pediatric hospital setting. The BioFire FA-RP2.1 test provides rapid results and can identify several viral and bacterial infections in a single test. This prospective, cross-sectional, diagnostic accuracy study enrolled participants ranging from 0 to 18 years of age, seeking medical consultation for any reason, who had been in contact with individuals confirmed to have COVID-19 or managed at the hospital for medical or surgical reasons. We employed a systematic sampling technique to ensure a representative sample. The study included 339 participants with a median age of 5 years. The BioFire FA-RP2.1 test detected SARS-CoV-2 in 18.6% of cases, while the reference RT-PCR test in 14% of cases. The BioFire FA-RP2.1 sensitivity and specificity for SARS CoV-2 detection were 98% and 94%, respectively. The positive probability coefficient (LR+) was 18. The agreement between the two tests was 0.80. In addition, the BioFire FA-RP2.1 test detected coinfection with two viruses in 7,6% of cases. The BioFire FA-RP2.1 is a reliable solution to meet pediatric healthcare needs and improve prognosis in the post-pandemic era thanks to its friendly interface and rapid testing process.

## Background

The novel severe acute respiratory syndrome coronavirus 2 (SARS-CoV-2) emerged in Wuhan, China, in December 2019, causing a significant public health crisis. Patients initially exhibited symptoms of severe pneumonia [1], which led to the naming of the disease as

**Funding:** This study was financially supported by Consejo Nacional de Ciencia y Tecnología, CONACYT, Paraguay (grant number PINV 20-287, MNMR), https://www.conacyt.gov.py/. The funder had no role in study design, data collection and analysis, decision to publish, or preparation of the manuscript. There was no additional external funding received for this study.

**Competing interests:** The authors have declared that no competing interests exist.

Coronavirus Disease 2019 (COVID-19). The rapid and widespread transmission of COVID-19 led to the World Health Organization (WHO) declaring it a pandemic in January 2020 [2].

Initially, the impact of COVID-19 appeared to be primarily on adults. Early reports from China, including Wuhan city and other regions, indicated that children predominantly presented with milder forms or were asymptomatic [3, 4] However, the emergence of the systemic multi-inflammatory syndrome in children several weeks after being infected by SARS-CoV-2 or in contact with COVID-19 patients prompted a reassessment of the risks faced by the pediatric population during the pandemic [5].

Children infected with SARS-CoV-2 exhibit symptoms comparable to those seen in adults, including fever, dry cough, and shortness of breath. Most of these cases typically recover within 1 to 2 weeks. However, children requiring admission to the pediatric intensive care unit have been reported [6]. Several studies showed that severe COVID-19 cases arise through an uncontrolled inflammatory response that leads to a cytokine-like syndrome [7, 8], suggesting cytokines play a central role in the pathogenesis of this disease. Besides, there is a correlation between the clinical advancement of COVID-19 and elevated levels of cytokines [9].

During the pandemic, the primary focus of health authorities was to minimize the transmission of the virus, particularly among vulnerable populations. The most effective strategy involved the identification of positive cases and the implementation of isolation measures. Given the highly contagious nature of SARS-CoV-2, rapid diagnostic tests were crucial to prevent the virus from spreading within hospitals and communities, thereby mitigating its impact on a larger scale [10]. According to the WHO, the reverse transcription followed by the polymerase chain reaction (RT-PCR), which detects the viral genome, is considered the gold standard for diagnosing SARS-CoV-2 infections. This high-sensitivity method allows for large-scale screening of virus infections in the population. However, its time-consuming, labor-intensive nature and the need for well-trained personnel restrict its implementation to centralized laboratories [11]. There is a growing need to develop and implement point-of-care and rapid tests for SARS-CoV-2 detection to address these limitations. In response to the escalating pandemic, innovative techniques to expedite the diagnostic process without compromising sensitivity and specificity were developed [12–14]. Besides accelerating the diagnostic process, implementing point-of-care and rapid tests for respiratory viral and bacterial infection diagnoses proved invaluable in improving appropriate antimicrobial prescriptions and optimizing therapy duration [15]. The Film-Array Respiratory panel (BioFire Diagnostics, Inc., Salt Lake City, Utah) is among these advancements. It is a multiplex real-time polymerase chain reaction test utilized during the past decade to detect viruses and bacteria in clinical samples. Its application in testing hospitalized patients for respiratory viruses has been linked to decreased healthcare resource utilization, including reduced usage of antibiotics and chest radiographs while promoting the increased implementation of isolation precautions. The Film Array Respiratory panel allowed for reduced lead time, waiting time, and turnaround time, as well as shorter hospital stays for pediatric patients [16, 17]. During the pandemic, the Film Array Respiratory panel incorporated the SARS-CoV-2 detection, resulting in the BioFire Film Array Respiratory Panel 2.1 (BioFire FA-RP2.1). This updated panel exhibited comparable performance to high-throughput RT-PCR assays for SARS-CoV-2 detection [18].

The General Pediatric Hospital "Children of Acosta Ñu" (GPHCAC) is a tertiary pediatric public hospital that provides care for nearly 400,000 children annually. It is a referral center for pediatric cardiac and bone marrow transplantation in the country. The Pediatric Emergency Department (PED) attends to approximately 120,000 children yearly. However, the SARS-CoV-2 pandemic had a profound impact on these numbers. Consultations and hospitalization rates experienced a decline of up to 80% and 50%, respectively, primarily due to the quarantine measures in the country aimed at controlling the virus spread [19].

The GPHCAC established a contingency facility in response to the evolving COVID-19 pandemic in the first semester of 2020. In line with the guidelines provided by health authorities, it was mandatory for all children visiting the hospital to undergo a SARS-CoV-2 test as part of the consultation process. The hospital utilized its Molecular Biology Laboratory, in operation since 2019 for diagnosing several viral infections, to perform the SARS-CoV-2 test. Initially, the hospital implemented a WHO-recommended, commercially available assay, the Light Mix Modular SarbeCoV E-gene EAV kit (TIB Molbiol, Berlin, Germany), for performing RT-PCR testing to detect SARS-CoV-2 infection. The Molecular Biology Laboratory tested over 4,000 nasopharyngeal samples for SARS-CoV-2 detection using this kit from January to March 2021. This test required an additional step for total RNA purification, extending the overall testing time to approximately 6 hours. The prolonged duration of this test was a considerable limitation, potentially delaying patient management and appropriate infection control measures. Recognizing the need for a more efficient testing method, we explored alternative options to expedite the diagnostic process and enhance patient care. This study aimed to assess the diagnostic accuracy of the BioFire FA-RP2.1 for SARS-CoV-2 detection at a pediatric reference hospital. The test demonstrated high sensitivity for SARS-CoV-2 detection in pediatric patients and proved highly effective in identifying other viral species responsible for infections within this population.

## Material and methods

### Study design

This prospective cross-sectional diagnostic accuracy study was conducted from April to August 2021. The study population included children aged 0–18 years managed at the contingency Pediatric Emergency Department (PED) of the GPHCAC. The eligible participants were children seeking medical consultation for any reason, who had been in contact with individuals confirmed to have COVID-19, or those managed at the hospital for medical or surgical reasons. Our inclusion criteria stipulated the requirement of obtaining informed consent from parents or guardians, while patients for whom BioFire FA-RP 2.1 analysis could not be conducted were excluded from the study. To evaluate the accuracy of the BioFire FA-RP 2.1 Panel (BioFire Diagnostics, Salt Lake City, UT, USA), our index test, we utilized the Light Mix Modular SarbeCoV E-gene EAV kit (TIB Molbiol, Berlin, Germany) as the reference test for comparison. Molbiol, Berlin, Germany) as the reference test for comparison.

The sample size for this study was determined considering the reported prevalence of SARS-CoV-2 infection cases in the pediatric population of Paraguay. According to the Ministry of Public Health, this prevalence was 8% [20]. The study employed the GRANMO software (Institut Municipal d'Investigació Mèdica, Barcelona, Spain) for sample size calculations, ensuring a statistical power of 90% at a significance level (α) of 0.05 and a type II error rate (β) of 0.10. The calculations considered a potential loss of 5% of samples, indicating that recruiting 328 children would provide a sufficient sample size for our study. This sample size enables us to detect a difference of 5.5% in the prevalence of SARS-CoV-2 compared to the reference value of 8%. Systematic sampling was employed in this study to select participants. The sampling process followed a periodic pattern using a sampling fraction. The sampling fraction was determined by dividing the total number of children who had undergone SARS-CoV-2 testing at the hospital two months before the study (n = 4920) by the desired sample size (n = 328). This calculation resulted in a sampling fraction of 15. Therefore, every 15th child tested for SARS-CoV-2 infection in chronological order was included in the study until reaching the required sample size. In cases where parents or guardians declined participation for a child, the recruitment sequence continued with the subsequent child. This process allowed for the

inclusion of eligible participants while maintaining a systematic and representative sampling approach.

## Nasopharyngeal samples

Clinical specimens were obtained from the participants by skilled healthcare personnel using nasopharyngeal swabs (Medico Technology Co. Ltd., China). The swab was inserted into each nostril until it reached the rhinopharynx, at approximately the distance from the nostrils to the external auditory canal. The swab was then gently left in contact with the mucosa for 5 seconds before being slowly removed using rotating movements. Subsequently, it was introduced into a sterile tube containing 3mL of the viral transport medium (Shenzhen Uni-Medica Technology Co. Ltd., China). The tube samples were transported to the Molecular Biology Laboratory following biosafety transport protocols for SARS-CoV-2 detection. These samples were an integral part of the routine RT-PCR reference test. A subset of clinical samples from the enrolled children underwent analysis using the BioFire Film Array Respiratory Panel 2.1 (BioFire Diagnostics, Salt Lake City, UT, USA) test. Trained personnel, distinct from those responsible for conducting the reference test, performed the BioFire FA-RP2.1 tests. Experienced professionals performed the analysis using blinded samples, ensuring a stringent and impartial methodology.

## The reference test for SARS-CoV-2 detection

An aliquot of 300μL of the viral transport media was utilized to extract total nucleic acids using the automated MagNA Pure LC 2.0 extraction system (Roche Diagnostic Ltd, Forrenstrasse, Switzerland), following the manufacturer's instructions. The total RNA was eluted from the purification column using 50μL of elution buffer and stored at -80˚C until further analysis. The nasopharyngeal samples were searched for SARS-CoV-2 RNA with the Light Mix Modular SarbeCoV E-gene EAV kit (TIB Molbiol, Berlin, Germany) and the LightCycler Multiplex RNA Virus Master (Roche Basel, Switzerland) on the LightCycler 480 or Cobas z 480 instruments (Roche Diagnostic Ltd., Forrenstrasse, Switzerland), following the manufacturer´s recommendations. The thermal cycling program consisted of an initial incubation at 55˚C for 3 minutes, followed by denaturation at 95˚C for 30 seconds and 45 cycles of denaturation at 95˚C for 5 seconds and annealing/extension at 60˚C for 15 seconds. The entire process, including nucleic acid purification and genomic amplification, takes approximately 6 hours to complete.

## The index test for SARS-CoV-2 detection

The index test for SARS-CoV-2 detection was the BioFire Respiratory Panel 2.1 (BioFire Defense LLC and BioFire Diagnostics LLC; Salt Lake City, UT, USA) run in the BioFire Film Array Torch equipment (BioFire Defense LLC and BioFire Diagnostics LLC; Salt Lake City, UT, USA), following the manufacturer's specifications. The BioFire RP 2.1 is a multiplex PCR-based test designed to identify various respiratory pathogens: Adenovirus (ADV), SARS-CoV-2, Coronaviruses 229E, HKU1, NL63, and OC43, Human Metapneumovirus, Human Rhinovirus/Enterovirus (HREV), Influenza A (including subtypes H1, H3, and H1-2009), Influenza B, Parainfluenza Virus, Respiratory Syncytial Virus, *Bordetella parapertussis*, *Bordetella pertussis*, *Chlamydia pneumoniae*, and *Mycoplasma pneumoniae*. The BioFire RP 2.1-EZ pouch, a disposable closed system, was utilized for this purpose. It contains all the necessary reagents for sample preparation, reverse transcription, polymerase chain reaction (PCR), and detection. The process involved combining the hydration solution and sample with BioFire FilmArray Sample Buffer and injecting them into the pouch. This pounch was inserted into the

instrument system and initiated the run. Automatic analysis of the endpoint DNA melting curve provided the result for each target assay, identifying the detected pathogen agent.

## Ethics statement

This study adhered to the principles outlined in the Declaration of Helsinki, ensuring ethical conduct throughout the research process. In light of the COVID-19 pandemic and the strict regulations mandated by health authorities in the country, we provided the parents or guardians of the children with a thorough explanation of the written informed consent. Those who received a verbal agreement from their parents or guardians participated in this study. The oral agreement was prioritized for the safety and well-being of the participants, reducing the risk of contact with potentially contaminated materials.

Participants' data records were thoroughly de-identified, ensuring complete anonymity to protect privacy and confidentiality. The Ethical Committee of the GPHCAC (IRB 00006311, number provided by the Office for Human Research Protection -OHRP-EEUU) reviewed and approved this study (approval number 000268).

## Statistical analysis

Patient data were analyzed using IBM's SPSS software. Quantitative variables were summarized by reporting their medians along with quartiles, providing information about the central tendency and variability of the data. On the other hand, qualitative variables were presented as percentages, indicating the distribution of participants across different categories.

To evaluate the diagnostic test's performance, including sensitivity, specificity, positive predictive value, negative predictive value, and positive and negative likelihood ratios, we utilized the Diagnosis Test Calculator from the University of Illinois at Chicago (http://ulan.mede.uic.edu/cgibin/testcalc). All statistical analyses were two-tailed.

## Results

### Characteristics of the study population

From April to August 2021, 5,063 children who received medical care at the GPHCAC were requested to undergo testing to detect SARS-CoV-2 infection, representing the eligible population. Through systematic sampling, we recruited 350 children for the study. However, the final participant number in the study was 339 children because the parents or guardians of 11 of them did not consent to their participation (Fig 1).

The participants had a median age of 5 years (Q1-1, Q3-11.2), with 51% being female. Hospitalized participants represented 60.8% of the children, while those receiving outpatient care were 39.2% of them. Most of the participants (90.5%) were symptomatic, exhibiting various types of symptoms. Suspected of having SARS-CoV-2 infection were 46% of the participants. Children with comorbidities accounted for 25.4% of the cases. Table 1 shows the complete demographic and clinical characteristics of the study population.

### Performance of the BioFire FA-RP 2.1 test

The BioFire FA-RP 2.1 detected SARS-CoV-2 infection in 63 out of 339 (18.5%) nasopharyngeal swab samples obtained from the participants. The reference RT-PCR test detected the virus in 14% (48/339) cases. Furthermore, the SARS-CoV-2 infection prevalence detected by both tests remained similar across all analyzed patient characteristics (Table 2).

The BioFire FA-RP 2.1 sensitivity was 95%, the specificity 94%, the positive predictive value 75%, and the negative predictive value 99%. The positive likelihood ratio (LH+) was 18 (IC

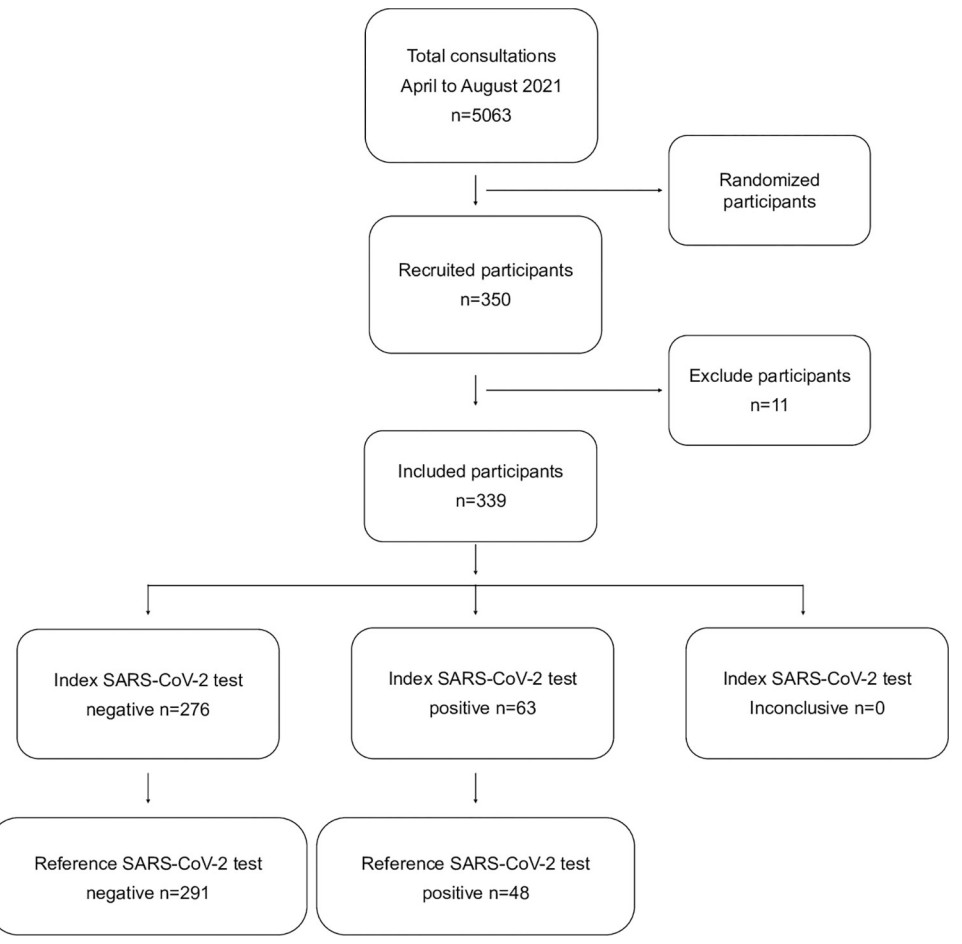

**Fig 1. Diagram showing the flow of participants through the study.**

95% 64–83), and the negative likelihood ratio was 0 (IC 95% 0–0.02) (Table 3). The agreement of the two tests was 0.80 (Kappa agreement r = 0.8).

Our analysis revealed that patients who tested positive for the reference RT-PCR test had a median age of 11 years (Q1 4 –Q3 15), whereas those who tested negative had a median age of 4 years (Q1 0.8 –Q3 10); p = 0,001 U (Mann Whitney test). Similarly, among the patients who tested positive for the BioFire FA-RP 2.1, 9 years was the median age (Q1 3 –Q3 14), while those who tested negative had a median age of 4 years (Q1 0.8 –Q3 10); p = 0,001 U (Mann Whitney test).

## Pathogens detected with the BioFire FA-RP 2.1 test

The BioFire FA-RP 2.1 test found viral infections in 115 participants. Among these cases, 26.5% (90/339) presented a single virus infection. The distribution of these single viral infections was as follows: SARS-CoV-2 in 42.2% (38/90), RSV (Respiratory Syncytial Virus) in 27.7% (25/90) Rhinovirus/Enterovirus in 25.5% (23 /90), and Adenovirus in 4.4% (4 /90) of the cases. This test allowed the identification of coinfection with two viruses in 7.4% (25 out of 339) of the participants. The most common combination was SARS-CoV-2 plus Rhinovirus/ Enterovirus in 68% (17 out of 25) cases. No bacterial infections were detected using the BioFire FA-RP 2.1 test. Table 4 shows a detailed overview of our findings.

**Table 1. Demographic and clinical characteristics of the study population (n = 339).**

| Characteristics | |
|---|---|
| | **Median (Q1 -Q3)** |
| **Age (years)** | 5 (1–11,2) |
| **Gender** | **n (%)** |
| Female | 173 (51) |
| Male | 166 (49) |
| **Health care** | |
| Outpatients | 133 (39.2) |
| Inpatients | 206 (60.8) |
| **Symptoms status** | |
| Symptomatic | 307 (90.6) |
| Asymptomatic | 32 (9.4) |
| **Symptoms** | |
| Fever and respiratory symptoms | 78 (25.4) |
| Respiratory symptoms | 72 (23.5) |
| Fever and gastrointestinal symptoms | 27 (8.8) |
| Fever | 24 (7.8) |
| Gastrointestinal symptoms | 21 (6.8) |
| Fever, gastrointestinal, and respiratory symptoms | 20 (6.5) |
| Respiratory and gastrointestinal symptoms | 10 (3.3) |
| Others | 55 (17.9) |
| **Reason for nasopharyngeal sampling** | |
| Suspected COVID-19 | 156 (46) |
| Hospital protocol upon admission | 141 (41.6) |
| Contact with people with COVID-19 | 42 (12.4) |
| **Others** | |
| Presenting comorbidities | 86 (25.4) |
| Requiring laboratory studies | 223 (65.8) |
| Requiring image studies | 129 (38) |

## Discussion

Several studies demonstrated the high sensitivity and specificity of BioFire FA-RP 2.1 in the adult population [18, 21–23]. Here, we present the first prospective evaluation of the BioFire FA-RP 2.1 accuracy for SARS-CoV-2 detection in a pediatric population. We used the WHO-recommended Light Mix Modular SarbeCoV E-gene EAV kit as the reference test. This test is a commercially available RT-PCR assay widely recognized as a reference test in hospital settings due to its consistently high sensitivity for SARS-CoV-2 detection, as evidenced by previous studies [24–26]. Our findings demonstrated the high sensitivity of BioFire FA-RP 2.1 for SARS-CoV-2 detection in children, a high positive likelihood ratio (LR+), and excellent agreement with the reference test. The probability that a sick patient will have the positive test with the film array was 18 times more likely than a healthy patient will have the same result. Our study revealed that BioFireFA RP 2.1 exhibited a specificity of 94%, which is similar to previous studies [18, 21, 23]. However, a study comparing BioFireFA RP 2.1 with the MAScIR SARS-CoV-2 M Kit 2.0 for SARS-CoV-2, a conventional real-time RT-PCR, reported a sensitivity of 100% and a specificity of 79.2% [22]. The lack of widely standardized protocols may contribute to the observed variations in sensitivity and specificity across different studies. To effectively address this issue, it is crucial to compare the diagnostic method against virus isolation, the gold standard for diagnosing virus infections.

**Table 2. Performance comparison between the BioFire FA-RP 2.1 and the reference RT-PCR tests according to the patient characteristics (positive testes, n = 111).**

| Characteristics | Reference test n = 48 | Index test n = 63 | p-Value |
|---|---|---|---|
| | Median (Q1 -Q3) | Median (Q1 -Q3) | |
| **Age (years)** | 11 (4–15) | 9 (3–14) | 0.80[a] |
| **Gender** | **n (%)** | **n (%)** | |
| Female | 23 (47.9) | 33 (52.4) | 0,78[b] |
| Male | 25 (52.1) | 30 (47.6) | |
| **Health care** | | | |
| Outpatients | 38 (79.2) | 48 (76.2) | 0.88[b] |
| Inpatients | 10 (20,8) | 15 (23.8) | |
| **Symptoms status** | | | |
| Symptomatic | 47 (98) | 62 (98.4) | 1[c] |
| Asymptomatic | 1 (2) | 1 (1.6) | |
| **Symptoms** | n = 47 | n = 62 | |
| Fever | 2 (4.3) | 3 (4.8) | 1[c] |
| Fever and respiratory symptoms | 17 (36.2) | 23 (37) | 1[b] |
| Gastrointestinal symptoms | 0 (0) | 1 (1.6) | 1[c] |
| Respiratory and gastrointestinal symptoms | 1 (2) | 3 (4.8) | 0.63[c] |
| Respiratory symptoms | 15 (32) | 16 (25.9) | 0.64[b] |
| Fever and gastrointestinal symptoms | 5 (10.6) | 8 (13) | 0.95[b] |
| Fever, gastrointestinal, and respiratory symptoms | 2 (4.3) | 1 (1.6) | 0.57[c] |
| Others | 5 (10.6) | 7 (11.3) | 1[b] |
| **Reason for nasopharyngeal sampling** | | | |
| Suspected COVID-19 | 24 (50) | 34 (54) | 0.82[b] |
| Hospital protocol upon admission | 5 (10.4) | 6 (9.5) | 1[b] |
| Contact with people with COVID-19 | 19 (39.6) | 23 (35.5) | 0.89[b] |
| **Others** | | | |
| Presenting comorbidities | 11 (22.9) | 13 (20.6) | 0.95[b] |
| Requiring laboratory studies | 13 (27) | 19 (30) | 0.88[b] |
| Requiring image studies | 14 (29) | 16 (25.4) | 0.82[b] |

a = U Mann Whitney

b = χ2

c = Fisher´s exact test

**Table 3. Accuracy of the BioFire FA-RP 2.1 compared to the reference RT-PCR test.**

| Index BioFire FA-RP2.1 test | Reference RT- PCR test | | Total |
|---|---|---|---|
| | Positive | Negative | |
| Positive | 47 | 16 | 63 |
| Negative | 1 | 275 | 276 |
| Total | 48 | 291 | 339 |
| Sensitivity | 98% | | |
| Specificity | 94% | | |
| Positive Predictive Value | 75% | | |
| Negative Predictive Value | 99% | | |
| Likelihood Ratio + | 18 | | |
| Likelihood ratio - | 0,02 | | |

**Table 4. Viruses identified with the BioFire FA-RP 2.1 test (n = 115).**

| Virus | n (%) |
|---|---|
| **Single virus (n = 90)** | |
| SARS CoV-2 | 38 (42.2) |
| RSV | 25 (27.7) |
| Rhinovirus/Enterovirus | 23 (25.4) |
| Adenovirus | 4 (4.4) |
| **Coinfection (n = 25)** | |
| SARS CoV-2/Rhinovirus-Enterovirus | 17 (68) |
| SARS CoV-2/RSV | 4 (16) |
| SARS CoV-2/Adenovirus | 4 (16) |

The final clinical diagnoses were as follows: COVID-19 in 14.1% (48/339), bronchiolitis in 10.3% (35/339), acute gastroenteritis in 6.7% (23/339), and wheezing in 2.6% (9/339) of cases. Among the bronchiolitis cases, 25/35 were attributed to RSV, while 10/35 showed SARS CoV-2 and RSV coinfection.

Classical real-time RT-PCR tests, such as the Light Mix Modular SarbeCoV E-gene EAV kit utilized in our study, possess high-throughput diagnostic capacity but are typically limited to centralized laboratories. However, with the widespread and significant reduction of SARS-CoV-2 infections, there is an escalating demand for decentralized testing capabilities for rapid COVID-19 case identification. The BioFire FA-RP 2.1 test can serve as a solution in this regard. Its closed and autonomous system obviates the requirement for an additional RNA pre-purification step, streamlining the testing workflow. The BioFire FA-RP 2.1 automatically identifies the pathogen agent in the clinical sample, yielding immediate results, and boasts a rapid turnaround time of 45 minutes, faster than the conventional methods that can take over 4 hours. In addition to the BioFire FA-RP 2.1, several other rapid COVID-19 testing platforms have emerged, including the Cepheid Xpert Xpress SARS-CoV-2 [27], DiaSorin Simplexa [28, 29], and GenMark ePlex SARS-CoV-2 tests [30]. These platforms offer alternative solutions for the efficient and timely detection of COVID-19 cases.

In this study, more than half of the positive cases detected by the reference standard and the BioFireFA RP 2.1 occurred in patients with suspected COVID-19, followed by individuals who had contact with COVID-19 patients. The age of the overall positive patients was higher than that of the negative ones. Our findings align with a previous study, which demonstrated an association between increasing age in children and higher odds of a positive test result [31]. Older children who attended in-person classes due to the relaxation of quarantine measures at the time of the study appeared to be the most exposed. This circumstance may contribute to our observed findings. The positive rate (18.5%) observed with the BioFireFA RP 2.1 in our study was higher than the 8% reported by healthcare authorities. The progression of the pandemic may explain these disparity results. As the pandemic continues, the prevalence of COVID-19 among children has increased. Therefore, our study, performed at a later pandemic stage, was more likely to find a higher SARS-CoV-2 infection prevalence than that reported by the health authorities among the pediatric population.

The BioFire FA RP 2.1 can detect multiple pathogens in a single test, while the reference RT-PCR tests can find only one pathogen at a time. As a result, our study was able to identify children infected with not only SARS-CoV-2 but also RSV, Rhinovirus/Enterovirus, and Adenovirus. A previous study found an 8.8% overall respiratory pathogen coinfection rate among COVID-19 adult patients associated with a worsening disease outcome [32]. In this study, the percentage of coinfection among SARS-CoV-2 positive children was nearly identical to the 7.5% reported by Karaaslan and colleagues [33], but without association with severe illness.

Viral respiratory tract infections, particularly in young infants are among the most prevalent diseases in children [34]. However, the COVID-19 pandemic has brought about a shift in respiratory epidemiology, with a decrease in respiratory infection incidence in children under five years old [35], linked provably to reduced social and educational activities [36].

Here, we found the most common virus among the children was SARS-CoV-2, followed by RSV. After the relaxation of health measures following the COVID-19 pandemic, several countries have experienced outbreaks of Respiratory Syncytial Virus (RSV) [37].

Early detection of SARS-CoV-2, Influenza, and RSV is critical, especially among vulnerable populations like infants and older adults. These viruses often present with similar symptoms, making early identification essential for timely and targeted medical interventions. Moving forward into the post-pandemic era, it becomes crucial to determine the viral causes of respiratory diseases, focusing on these viruses. Commercially available multiplex PCR assays, such as BioFire FA RP 2.1 and other recently reported tests [36], enable point-of-care differential diagnosis of these viral infections. These advanced assays provide healthcare professionals with valuable information to make informed decisions and deliver appropriate care, ultimately contributing to better patient outcomes and effective management of viral outbreaks.

A limitation of this study arose from the restricted number of participating children. The main factor contributing to this limitation was the availability of the BioFire FA RP 2.1 kit during the ongoing pandemic. Nonetheless, given the prevalence of positive SARS-CoV-2 cases among children, the sample size retained adequate statistical power. Additionally, the patient population of this study represents a typical group of patients seeking hospital care.

## Conclusions

The BioFire FA RP2.1 test exhibited a high sensitivity compared to the reference RT-PCR test in a pediatric hospital setting. The positive probability coefficient (LR+) was 18, and there was good agreement with the reference test (r = 0.8). The test also detected coinfection with other viruses in 7.4% of the SARS-CoV-2 infected patients, which is relevant information for the attending pediatrician. This test is a reliable solution to meet pediatric healthcare needs and improve prognosis in the post-pandemic era thanks to its friendly interface and rapid testing process.

## Supporting information

**S1 Data.**
(XLSX)

## Acknowledgments

The authors thank Dr Pio Alfieri, CEO of the GPHCAC, Dr. Fernando Arevalos, Dr. Lady Franco, Dr. Claudia Marecos, and the Microbiology Laboratory staff of the General Pediatrics Hospital "Children of Acosta Ñu" for their support.

The authors also thank Dr. Susana Sanchez Bernal and Laura Mendoza, directors of SANU-PAR, the institution responsible for grant management.

## Author Contributions

**Conceptualization:** Mirta Mesquita Ramirez, Victor Hugo Aquino.

**Data curation:** Mirta Mesquita Ramirez.

**Formal analysis:** Mirta Mesquita Ramirez, Miria Noemi Zarate, Leonidas Adelaida Rodriguez, Victor Hugo Aquino.

**Funding acquisition:** Mirta Mesquita Ramirez.

**Investigation:** Miria Noemi Zarate, Leonidas Adelaida Rodriguez.

**Methodology:** Miria Noemi Zarate, Leonidas Adelaida Rodriguez.

**Writing – original draft:** Mirta Mesquita Ramirez, Victor Hugo Aquino.

**Writing – review & editing:** Mirta Mesquita Ramirez, Victor Hugo Aquino.

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
