## [Decision Letter · Decision Letter 0]

4 Sep 2023

PONE-D-23-25121Performance evaluation of Biofire Film Array Respiratory Panel 2.1 for SARS-CoV-2 detection in a pediatric hospital settingPLOS ONE

Dear Dr. Aquino,

Thank you for submitting your manuscript to PLOS ONE. After careful consideration, we feel that it has merit but does not fully meet PLOS ONE’s publication criteria as it currently stands. Therefore, we invite you to submit a revised version of the manuscript that addresses the points raised during the review process.

We look forward to receiving your revised manuscript.

Kind regards,

Benjamin M. Liu, MBBS, PhD, D(ABMM), MB(ASCP)

Academic Editor

PLOS ONE

Journal Requirements:

This study was financially supported by Consejo Nacional de Ciencia y Tecnología, CONACYT, Paraguay (grant number  PINV 20-287), https://www.conacyt.gov.py/. The funder had no role in study design, data collection and analysis, decision to publish, or preparation of the manuscript.  

3.In your Data Availability statement, you have not specified where the minimal data set underlying the results described in your manuscript can be found. PLOS defines a study's minimal data set as the underlying data used to reach the conclusions drawn in the manuscript and any additional data required to replicate the reported study findings in their entirety. All PLOS journals require that the minimal data set be made fully available. For more information about our data policy, please see http://journals.plos.org/plosone/s/data-availability.

Reviewers' comments:

Reviewer's Responses to Questions

**Comments to the Author**

1. Is the manuscript technically sound, and do the data support the conclusions?

Reviewer #1: Yes

2. Has the statistical analysis been performed appropriately and rigorously? 

Reviewer #1: Yes

3. Have the authors made all data underlying the findings in their manuscript fully available?

Reviewer #1: Yes

4. Is the manuscript presented in an intelligible fashion and written in standard English?

Reviewer #1: Yes

5. Review Comments to the Author

Reviewer #1: This is a useful study demonstrating the clinical utility of BioFire Respiratory panel for the detection of SARS-CoV-2. The study is well-performed and the paper is well written.

1. The authors should cite more extensively to introduce the pathogenesis and clinical significance based on the following references.

Liu BM, Martins TB, Peterson LK, Hill HR. Clinical significance of measuring serum cytokine levels as inflammatory biomarkers in adult and pediatric COVID-19 cases: A review. Cytokine. 2021 Jun;142:155478. doi: 10.1016/j.cyto.2021.155478. Epub 2021 Feb 23. PMID: 33667962; PMCID: PMC7901304.

Hu B, Huang S, Yin L. 2020. The cytokine storm and COVID-19. J Med Virol. DOI: 10.1002/jmv.26232.

Liu B, Hill HR. Role of Host Immune and Inflammatory Responses in COVID-19 Cases with Underlying Primary Immunodeficiency: A Review. J Interferon Cytokine Res. 2020 Dec;40(12):549-554. doi: 10.1089/jir.2020.0210. PMID: 33337932; PMCID: PMC7757688.

2. Please clarify inclusion and exclusion criteria

3. Is the statistical analysis two-tailed or one-tailed?

4. In the Introduction or Discussion, please discuss the following point based on the following reference: though the BioFire Respiratory panel is a large multiplex PCR panel, it still lacks some important respiratory pathogen to be covered, e.g., Pneumocystis jirovecii, which can be detected otherwise by some other novel molecular assays.

Liu B, Totten M, Nematollahi S, Datta K, Memon W, Marimuthu S, Wolf LA, Carroll KC, Zhang SX. Development and Evaluation of a Fully Automated Molecular Assay Targeting the Mitochondrial Small Subunit rRNA Gene for the Detection of Pneumocystis jirovecii in Bronchoalveolar Lavage Fluid Specimens. J Mol Diagn. 2020 Dec;22(12):1482-1493. doi: 10.1016/j.jmoldx.2020.10.003. Epub 2020 Oct 15. Erratum in: J Mol Diagn. 2021 Apr;23(4):506. PMID: 33069878.

5. In the Introduction or Discussion, please discuss that BioFire Respiratory panel provides some level of commutablity and standardization in different clinical labs as the same platform and panel is being used. According to the following reference, different reverse transcriptase with varied efficiency may cause unreliable results. BioFire Respiratory panel solves this problem as a popular and widely used commercial molecular test. Please discuss this based on this reference.

Liu B, Forman M, Valsamakis A. Optimization and evaluation of a novel real-time RT-PCR test for detection of parechovirus in cerebrospinal fluid. J Virol Methods. 2019 Oct;272:113690. doi: 10.1016/j.jviromet.2019.113690. Epub 2019 Jul 5. PMID: 31283959.

6. Error: Line 57: "Wuhan province" should be "Wuhan city"

6. PLOS authors have the option to publish the peer review history of their article (what does this mean?). If published, this will include your full peer review and any attached files.

Reviewer #1: No

---

## [Author Response · Author response to Decision Letter 0]

15 Sep 2023

Response to Editor and Reviewers

We appreciate the constructive feedback and guidance from the editor, as well as the comments from the reviewers. We are confident that these constructive inputs will significantly enhance the quality of our manuscript.

1. When submitting your revision, we need you to address these additional requirements. Please ensure that your manuscript meets PLOS ONE's style requirements, including those for file naming. The PLOS ONE style templates can be found at https://journals.plos.org/plosone/s/file?id=wjVg/PLOSOne_formatting_sample_main_body.pd f and https://journals.plos.org/plosone/s/file?id=ba62/PLOSOne_formatting_sample_title_authors_ affiliations.pdf

Answer

We have carefully reviewed the additional requirements and have made the necessary revisions to ensure that our manuscript complies with PLOS ONE's style requirements, including file naming. We have used the provided PLOS ONE style templates to guide us in this process.

This study was financially supported by Consejo Nacional de Ciencia y Tecnología, CONACYT, Paraguay (grant number PINV 20-287), https://www.conacyt.gov.py/. The funder had no role in study design, data collection and analysis, decision to publish, or preparation of the manuscript.

Answer

We provided an amended statement that declares all the funding or sources of support received during this study. We also included the statement 'There was no additional external funding received for this study' in our updated Funding Statement within our cover letter.

 Upon re-submitting your revised manuscript, please upload your study’s minimal underlying data set as either Supporting Information files or to a stable, public repository and include the relevant URLs, DOIs, or accession numbers within your revised cover letter. For a list of acceptable repositories, please see http://journals.plos.org/ plosone/s/data-availability#loc-recommended-repositories. Any potentially identifying patient information must be fully anonymized.

Answer

In the Data Availability Statement, we have mentioned that “All relevant data are within the manuscript and its Supporting Information files”. Additionally, we have uploaded as a supporting file the Data Base including anonymized participants' demographic, clinical, and laboratory characteristics, where the minimal data set underlying the results described in this manuscript can be found.

Answer

References were reviewed and formatted according to the Plos One journal requirements.

Reviewer #1:

This is a useful study demonstrating the clinical utility of BioFire Respiratory panel for the detection of SARS-CoV-2. The study is well-performed and the paper is well written.

Thank you for your thoughtful review and positive feedback regarding our study. We appreciate your valuable comments.

1. The authors should cite more extensively to introduce the pathogenesis and clinical significance based on the following references.

Liu BM, Martins TB, Peterson LK, Hill HR. Clinical significance of measuring serum cytokine levels as inflammatory biomarkers in adult and pediatric COVID-19 cases: A review. Cytokine. 2021 Jun;142:155478. doi: 10.1016/j.cyto.2021.155478. Epub 2021 Feb 23. PMID: 33667962; PMCID: PMC7901304.

 Hu B, Huang S, Yin L. 2020. The cytokine storm and COVID-19. J Med Virol. DOI: 10.1002/jmv.26232.

Liu B, Hill HR. Role of Host Immune and Inflammatory Responses in COVID-19 Cases with Underlying Primary Immunodeficiency: A Review. J Interferon Cytokine Res. 2020 Dec;40(12):549-554. doi: 10.1089/jir.2020.0210. PMID: 33337932; PMCID: PMC7757688.

Answer

We concur with the significance of introducing the role of cytokines in the pathogenesis of COVID-19 within the introduction section. We have emphasized the great relevance of this matter and integrated the suggested references (lines 66-71).

2. Please clarify inclusion and exclusion criteria

Answer

The inclusion and exclusion criteria were clarified in the text (lines 142-144).

3. Is the statistical analysis two-tailed or one-tailed?

Answer

All statistical analyses were conducted using a two-tailed approach. This clarification has been included in the text for clarity (lines 246-247).

4. In the Introduction or Discussion, please discuss the following point based on the following reference: though the BioFire Respiratory panel is a large multiplex PCR panel, it still lacks some important respiratory pathogen to be covered, e.g., Pneumocystis jirovecii, which can be detected otherwise by some other novel molecular assays.

Liu B, Totten M, Nematollahi S, Datta K, Memon W, Marimuthu S, Wolf LA, Carroll KC, Zhang SX. Development and Evaluation of a Fully Automated Molecular Assay Targeting the Mitochondrial Small Subunit rRNA Gene for the Detection of Pneumocystis jirovecii in Bronchoalveolar Lavage Fluid Specimens. J Mol Diagn. 2020 Dec;22(12):1482- 1493. doi: 10.1016/j.jmoldx.2020.10.003. Epub 2020 Oct 15. Erratum in: J Mol Diagn. 2021 Apr;23(4):506. PMID: 33069878.

5. In the Introduction or Discussion, please discuss that BioFire Respiratory panel provides some level of commutablity and standardization in different clinical labs as the same platform and panel is being used. According to the following reference, different reverse transcriptase with varied efficiency may cause unreliable results. BioFire Respiratory panel solves this problem as a popular and widely used commercial molecular test. Please discuss this based on this reference.

Liu B, Forman M, Valsamakis A. Optimization and evaluation of a novel real-time RT-PCR test for detection of parechovirus in cerebrospinal fluid. J Virol Methods. 2019 Oct;272:113690. doi: 10.1016/j.jviromet.2019.113690. Epub 2019 Jul 5. PMID: 31283959.

Answer

In response to these two comments (4 and 5), we respectfully disagree with the reviewer regarding the relevance of incorporating the suggested references into our study discussion. The first reference pertains to the development of an assay for the detection of

Pneumocystis jirovecii in Bronchoalveolar samples, a topic not within the scope of our study, which exclusively evaluates the BioFire Respiratory panel for SARS-CoV-2 detection. Similarly, the second reference concerns parechovirus detection, which is unrelated to the subject of our study.

Consequently, we believe these references do not align with the focus of our research and, therefore, cannot be reasonably discussed in our study.

6. Error: Line 57: "Wuhan province" should be "Wuhan city"

Answer

The correction has been implemented.

Additional clarification

We would like to clarify that there was a typographical error in our previous statement regarding the BioFire FA-RP2.1 specificity. The accurate specificity for this test is 94%, not 75%. As a result, we have made the necessary revisions to the discussion section (lines 327-330).

---

## [Editor Report · Decision Letter 1]

18 Sep 2023

Performance evaluation of Biofire Film Array Respiratory Panel 2.1 for SARS-CoV-2 detection in a pediatric hospital setting

PONE-D-23-25121R1

Dear Dr. Aquino,

We’re pleased to inform you that your manuscript has been judged scientifically suitable for publication and will be formally accepted for publication once it meets all outstanding technical requirements.

Kind regards,

Benjamin M. Liu, MBBS, PhD, D(ABMM), MB(ASCP)

Academic Editor

PLOS ONE
---

## [Editor Report · Acceptance letter]

28 Sep 2023

PONE-D-23-25121R1 

Performance evaluation of Biofire Film Array Respiratory Panel 2.1 for SARS-CoV-2 detection in a pediatric hospital setting 

Dear Dr. Aquino:

I'm pleased to inform you that your manuscript has been deemed suitable for publication in PLOS ONE. Congratulations! Your manuscript is now with our production department. 

Kind regards, 

on behalf of

Dr. Benjamin M. Liu 

Academic Editor

PLOS ONE